# *Bacillus subtilis* Effects on Growth Performance and Health Status of *Totoaba macdonaldi* Fed with High Levels of Soy Protein Concentrate

**DOI:** 10.3390/ani12233422

**Published:** 2022-12-05

**Authors:** Jorge Olmos, Lus M. López, Antonio Gorriño, Mario A. Galaviz, Victor Mercado

**Affiliations:** 1Molecular Microbiology Laboratory, Department of Marine Biotechnology, Centro de Investigación Científica Y de Educación Superior de Ensenada, Ensenada 22860, Mexico; 2Facultad de Ciencias Marinas, Universidad Autónoma de Baja California (UABC), Carretera Transpeninsular Ensenada–Tijuana No. 3917, Col. Playitas, Ensenada 22860, Mexico

**Keywords:** totoaba, soybean, *Bacillus*, probiotic, functional feeds

## Abstract

**Simple Summary:**

In this study, we investigated *Bacillus subtilis* 9b effects in *Totoaba macdonaldi* fed with 30% and 60% of soy protein concentrate substitution. We found that *B. subtilis* 9b supplementation improved feed intake, weight gain, and internal organs condition of *T. macdonaldi* fed with 30% substitution. Animals fed with 60% of SPC substitution and *B. subtilis* doubled their weight and presented 20% more survival than its control diet without *B. subtilis* 9b probiotic strain. *B. subtilis* 9b was able to modulate *T. macdonaldi* intestinal microbiota and increase its resistance to *Vibrio harveyi* pathogenic strain.

**Abstract:**

*T. macdonaldi* is a carnivorous species endemic to the Gulf of California. Indiscriminate exploitation has put totoaba at risk, inducing the development of aquaculture procedures to grow it without affecting the wild population. However, aquafeeds increasing cost and low yields obtained with commercial feeds have motivated researchers to look for more nutritious and cheaper alternatives. Soybean (SB) is the most popular alternative to fishmeal (FM); however, antinutritional factors limit its use in carnivorous species. In this study, we analyzed *B. subtilis* 9b probiotic capacity to improve growth performance and health status of *T. macdonaldi* fed with formulations containing 30% and 60% substitution of fish meal with soy protein concentrate (SPC). In addition, we investigated its effect on internal organs condition, their capacity to modulate the intestinal microbiota, and to boost the immunological response of *T. macdonaldi* against *V. harveyi* infections. In this sense, we found that *T. macdonaldi* fed with SPC30Pro diet supplemented with *B. subtilis* 9b strain and 30% SPC produced better results than SPC30C control diet without *B. subtilis* and DCML commercial diet. Additionally, animals fed with SPC60Pro diet supplemented with *B. subtilis* 9b strain and 60% SPC doubled their weight and produced 20% more survival than SPC60C control diet without *B. subtilis*. Thus, *B. subtilis* 9b improved *T. macdonaldi* growth performance, health status, modulated intestinal microbiota, and increased animal’s resistance to *V. harveyi* infections, placing this bacterium as an excellent candidate to produce functional feeds with high levels of SPC.

## 1. Introduction

Aquaculture is one of the fastest-growing industries due to the high demand for aquatic food products. However, this industry requires a fast adaptation to increasing problems such as aquafeed (feed for farmed aquatic animals) ingredients price, diseases, and low yields [1]. Fish products are the main source of proteins and lipids in aquafeed formulations; the balanced amount of essential amino acids and fatty acids required for optimum development, growth, and reproduction of aquacultured animals is of great importance [2]. However, overexploitation of small pelagic fishes to produce aquafeeds and livestock feeds has induced ecological problems and increased the price of fish products. Therefore, research turned to economic, more accessible, and sustainable alternatives to aquafeeds formulation [3].

In this sense, feeds that provide health benefits beyond their nutritional value known as functional feed (FF) seem to be an economically attractive, environmentally friendly, and safe alternative to improve growth performance and health status and induce physiological benefits beyond traditional feeds in cultivated organisms [4]. Alternatives to substitute fish products have been studied throughout the years; poultry, livestock, and fish byproducts, as well as plant and unicellular products are among the most important [5]. Due to their low cost and abundance, vegetable ingredients have become the most investigated to substitute fish products in aquafeeds formulation [3,5,6]. Soybean products (SBP) are of great interest in animal feeding due to their high protein and lipid content, making them an excellent alternative to fish products in aquafeeds production [5,6]. However, antinutritional factors (ANF) such as non-digestible carbohydrates, enzyme inhibitors, and allergenic proteins contained in soybean meal (SBM) and SPC, have limited the use of SBP in aquafeeds production. ANF can induce health problems such as distal intestine enteritis, pancreatic hypertrophy, liver damage, and animal’s death [7,8,9,10,11]. Despite the adverse effects of ANF, SBP have been included in feeds of aquacultured animals such as *Litopenaeus vannamei* [12], *Oncorhynchus mykiss* [13], *Diplodus puntazzo* [14], *Salmo salar* [15], and *Siganus rivalatus* [16]. Carnivore fishes such as *Totoaba macdonaldi* have shown less tolerance to SBM and SPC due to the lack of enzymes capable of digesting ANF present in the SBP [17]. Therefore, the addition of probiotic bacteria in aquafeeds to facilitate SBP digestion and assimilation could be a sustainable, safe, and economical solution for the aquaculture industry.

*Bacillus subtilis* is a spore-forming bacteria with the capacity to increase the assimilation of vegetable ingredients by producing exogenous enzymes such as proteases, lipases, and carbohydrases. This bacterium can inhibit pathogen development and stimulate the host’s immune system by producing secondary metabolites such as bacteriocin, and modulating the gut microbiota [18,19,20]. *B. subtilis* has been classified as GRAS by the Food and Drug Administration (FDA), making it safe to use in humans and animals. The addition of *Bacillus* in animals fed with SBP has shown growth and health improvement in *Epinephelus coioides* [21], *L. vannamei* [22,23], and *Lithobates catesbeianus* [24]. Thus, implementing *Bacillus* probiotic strains to feed economically attractive fishes such as *T. macdonaldi* could help solve problems caused by SBP.

*T. macdonaldi* is a carnivorous species endemic to the Gulf of California and is considered the largest member of the *Sciaenidae* family [25]. This fish reduced its population in the last decades due to sport and illegal fishing [26]. By 1986, *T. macdonaldi* was included in the list of endangered species by the IUCN Red List of Threatened Species [27]. In the 1990s, research began to establish the bases of its cultivation and understanding its reproductive cycle in captivity [28]. Results in *T. macdonaldi* production have attracted the attention of the aquaculture industry [29]. Therefore, improving feed assimilation and feed price optimization are required to increase productivity and sustainability.

*T. macdonaldi* tolerates around 20% SBM and 30% SPC substitution without suffering severe health problems [30,31,32,33]. Thus, implementing probiotics such as *B. subtilis* in *T. macdonaldi* aquafeeds could improve tolerance and reduce adverse effects induced by SBP. Therefore, this study focused on *Bacillus subtilis* 9b strain addition in *T. macdonaldi* feeds formulated with 30% and 60% of FM substitution by SPC and evaluated its effect on growth performance, health status, and intestinal microbiota modulation.

## 2. Materials and Methods

### 2.1. Feeds Preparation

*B. subtilis* 9b was obtained from the molecular microbiology laboratory at Ensenada Center for Scientific Research and Higher Education (CICESE; Ensenada, Baja California, Mexico). The enzymatic capacity of this strain was described previously by Ochoa-Solano and Olmos-Soto [20]. Four isonitrogenous (42%) and isolipidic (7%) feeds were formulated to contain 30% (SPC30C; SPC30Pro), and 60% (SPC60C; SPC60Pro) FM substitution with SPC (Table 1). DCML feed from Skretting© (Product Number 12143085) was used as commercial control (Crude Protein 46%, Crude Fat 12%, Ash 12%, Moisture 8%, Crude Fiber 3%, Phosphorus 1.5% NFE 17%, and Sodium 1%). For feed preparation, macro ingredients were thoroughly mixed, vitamins and minerals were then incorporated gradually. Oil-based ingredients and water were added to obtain the final consistency. *B. subtilis* 9b strain was supplemented at 0.1% (1 g/kg) before pellets formation and drying at 60 °C for 24 h.

### 2.2. Organisms and Growth Conditions

Juveniles of *T. macdonaldi* were provided by the marine finfish hatchery of Facultad de Ciencias Marinas, FCM, Universidad Autónoma de Baja California (Ensenada, Baja California, México). Fish were fed with a feed containing 46% protein and 12% lipids and acclimated to experimental facility for two weeks at FCM. A total of 180 fish with an average weight of 153 ± 0.84 g were randomly selected and stocked in 15 tanks of 1100 L. Tanks with 12 fish were connected to a water recirculation system and the flow rate was adjusted to allow 10% of water exchange per day per tank. Water parameters were monitored daily to maintain stable experimental conditions. Temperature was kept at 25 ± 1 °C with thermo-controlled chillers. Salinity was measured with a refractometer and maintained at 35.0 ± 0.5%_o_. Photoperiod was kept at 12:12 h light:dark. Oxygen concentration was measured with a YSI Pro 20l oximeter and kept above 6 mg/L. Total ammonia-nitrogen (NH_4_^+^-N) and total nitrite-nitrogen (NO_2_^−^-N) were measured daily before the first meal of the day with colorimetric test kits (Aquarium Pharmaceutical, Mars, PA, USA). The level of pH was measured using an Oakton pHTestr10 pH meter with an accuracy of 0.01 pH units.

Experiments were performed in triplicate, and fish were fed to apparent satiety twice a day (08:00 and 16:00 h), seven days a week for three months. Feed leftovers were re-weighted to measure feed intake. Experimental procedures related to fish husbandry were approved by the Secretariat of Agriculture, Livestock, Rural Development, Fisheries, and Food (Mexican Official Standard NOM-062-ZOO-1999).

### 2.3. Growth Performance

Fish were anesthetized using clove oil at 0.01% (*v*/*v*) and weighed individually at the beginning and end of the experiment (24 h fasting before body weight measurement). The following equations were used to evaluate fish growth parameters [22,30]:Weight gain (WG)
(1)$$WG=\left[\frac{FW-IW}{FIW}\right]*100$$
Specific growth rate (SGR)
(2)$$SGR=\left[\frac{LnFW-LnIW}{t}\right]*100$$
Thermal growth coefficient (TGC)
(3)$$TGC=\left[\frac{\sqrt[{3}]{FW}-\sqrt[{3}]{IW}}{\left(temperature\;{}^{\circ}C\right)\;x\;t\;}\right]*1000$$
Daily feed intake (DFI)
(4)$$DFI=\sum i\;n\;\left[\frac{total\;feed\;consumed\;\left(g\right)\;}{number\;of\;fish}\right]÷t$$
Survival rate
(5)$$S=\left[\frac{final\;No.\;fish\;}{initial\;No.\;fish}\right]*100$$
where: *FW* = Average final weight (g); *IW* = Average initial weight (g), *FIW*: Fish initial weight (g); *LnFW* = *Ln* final weight; *LnIW* = *Ln* initial weight; *t* = time (days).

### 2.4. Intestine Recollection and Organs Observation

At the end of the bioassay, two fish per treatment were overdosed with clove oil and sacrificed to observe visceral fat content, liver, spleen, distal intestine condition, fillet firmness by tactile and visual assessment and collect samples of the distal intestine to perform a fluorescent in situ hybridization procedure (FISH).

### 2.5. Fluorescent In Situ Hybridization

To analyze bacterial communities in the totoaba distal intestine, fluorescent in situ hybridization was performed following the methodology described by Hernandez & Olmos [34] with slight modifications in control and sample preparation.

#### 2.5.1. Probes

Probes for low G + C Gram-positive bacteria labeled with Carboxy-X-rhodamine [5-(6)-ROX] and *γ-Proteobacteria* labeled with isothiocyanate (5-FITC) previously described by Hernandez & Olmos were used [34].

#### 2.5.2. Control Preparation

Bacterial samples of *B. subtilis* 9b and *V. harveyi* (ATCC 14126) were used as low G + C and *γ-Proteobacteria* controls, respectively. The strains were grown in specialized media until they reached the mid-exponential growth phase at 600 nm. Samples were fixed by adding 37% (*v*/*v*) filtered formaldehyde to a final concentration of 6% (*v*/*v*) and incubated at 4 °C for 1 h. A total of 1 mL of cells were collected by centrifugation at 12,000× *g* for 3 min. Pellets were washed twice with ice-cold phosphate-buffered saline (PBS; 120 mM NaCl, 2.7 mM KCl, 10 mM phosphate, pH 7,4), suspended in 1 mL of PBS and stored at −20 °C for future use; no longer than 2 months.

#### 2.5.3. Samples Preparation

Intestine tissue samples were taken at the beginning and end of the experiment; 1 g of distal intestine was collected from five fish at the beginning of the study and from two per treatment at the end of the experiment, suspended in 5 mL of formaldehyde (10%) and mechanically homogenized. Aliquots of 500 µL were centrifuged at 19,800× *g* for 8 min; tissue pellets were washed twice with ice-cold PBS and suspended in 1 mL of PBS.

Samples and controls were serially diluted and 15 µL aliquots were spread onto 12-wells Teflon slides. Autofluorescence control was carried out by inoculating wells with reference strains without probes. Slides were allowed to dry at 37 °C, treated with ethanol/formaldehyde solution (90:10 *v*/*v*) for 5 min, rinsed twice in distilled water and dried again at 37 °C.

#### 2.5.4. Sample Fixation

In situ hybridization of the controls and tissue samples was performed as described previously [34]. Briefly, 40µL of hybridization mixture [10X SET buffer (1× SET: 150 mM NaCl, 20 mM Tris-HCl (pH 7,8), 1 mM EDTA), 0.2% bovine serum albumin (*w*/*v*), 0.01% polyadenylic acid (*w*/*v*), and 11% of dextran sulfate (*w*/*v*)] containing 1 ng/l of probes were added to samples immobilized on slides, avoiding bubble formation. The slides were placed into a humidified chamber with pieces of 1X SET-saturated paper and incubated overnight in the dark at 37 °C. After hybridization, the slides were washed twice with 1X SET buffer for 20 min each. Slides were dried at 37 °C in the dark, and each well was overlaid with 5 µL of mounting fluid (10× SET buffer, 50% glycerol, 0.1% p-phenylenediamine HCl). Finally, a cover slip was placed, and samples were analyzed.

#### 2.5.5. Microscopy

Fluorescence signals were analyzed using an Olympus microscope with MF filters (400–700 nm: 5FITC or 5(6)-ROX) and Plan Fluorite Universal objectives. The excitation source was a 50-W high-pressure mercury bulb. Digital images were processed with image Fiji/Image J software (version 1.52g, Java 1.80_172, U.S. National Institutes of Health, Bethesda, Maryland, USA).

Bacterial density from the distal intestine was estimated by digital image count, checked for normality, and when appropriate, transformed. For cell quantification, means were calculated from 10 randomly chosen fields for each sample [35].

#### 2.5.6. Bacterial Quantification by FISH

Quantification was conducted by considering the well area, area of the picture, and volume and dilution of the sample:(6)$$Bacteria/mL=\frac{\left(bacteria\;count*fiels\;per\;well\right)\left(1000\;µL\;PBS\right)}{15\;µL\;of\;dilution\;applied\;to\;the\;slide}$$
(7)$$Fields\;per\;well=\frac{well\;area}{\;picture\;area}$$
where: *well area* = 19,635,000 µm; *picture area* = 203,970 µm.

### 2.6. Pathogens Resistance Assay

After completing the feeding trials, six fish from DCML, SPC30C, and SPC30Pro treatments were placed in tanks isolated from the system and half-filled with water. *V. harveyi* CAIM1508 pathogenic strain was grown in specialized media until reaching stationary growth phase at 600 nm. An amount of 10 mL were centrifuged at 12,000× *g* for 10 min, washed twice with cold PBS, and suspended in PBS to obtain a final concentration of 10^7^ cells/mL. Clove oil was used as an anesthetic; fish were inoculated intraperitoneally with *V. harveyi* solution. Fish were placed in their tanks, and recovery was recorded by visualizing the swimming fish activity and feed intake post-infection.

### 2.7. Statistical Analysis

Collected data from growth performance were analyzed through a normality and homoscedasticity test. One way-analysis of variance (ANOVA) was performed to observe differences between treatments and their respective controls without probiotics. *p*-values < 0.05 were considered significant. Tukey’s post hoc test was applied for data presenting normality, and for those data that did not present normality, analysis of multiple comparisons by Kruskal–Wallis was applied.

## 3. Results

### 3.1. Feeding Trials

Table 2 describes the growth performance of fish fed with DCML, SPC30C, and SPC30Pro treatments, as well as with SPC60C and SPC60Pro. Results show 100% survival rate for diets with 30% substitution and 61.11% and 83.33% for diets with 60% substitution, respectively. Although fish fed with SPC30C and SPC30Pro do not show significant differences in their growth performance, SPC30Pro treatment supplemented with *B. subtilis* 9b probiotic strain produced higher FW, WG, SGR, TGC, and DFI values than SPC30C formulation (Table 2). Fish fed with 60% substitution diets presented lower growth performance than DCML and 30% substitution diets (Table 2); however, significant differences in most parameters were obtained when *B. subtilis* 9b probiotic strain was included in SPC60Pro formulation, compared to SPC60C without probiotic addition.

### 3.2. Treatment Effect on Organ Conditions

At the end of the feeding trial, three animals of each treatment were sacrificed, and their muscle condition, visceral fat content, as well as their liver, spleen, and distal intestine were measured and analyzed (Table 3).

Fish fed with SPC30C and SPC30Pro diets produced firmer fillets than fish fed with DCLM, SPC60C, and SPC60Pro. In this sense, fish fed with SPC60C diet showed worse muscular conditions in the study; however, fish fed with SPC60Pro produced muscular conditions similar to animals fed with DCML commercial diet (Table 3).

The visceral fat content was lower in diets with 30 and 60% substitution than in DCML commercial diets. In addition, the liver, spleen, and intestine were larger in fish fed with 30% substitution diets than in 60% substitution diets and DCML commercial diets (Table 3). Moreover, bigger spleen and intestine can be observed in fish fed with SPC60Pro than in fish fed with SPC60C treatment without *B. subtilis* 9b strain (Table 3).

### 3.3. Fluorescent In Situ Hybridization Evaluation

Intestinal samples were taken from fish at the beginning and end of the feeding trial to evaluate the capacity of *B. subtilis* 9b strain to modulate the intestinal microbiota; increasing low G + C Gram-positive bacteria and reducing *γ-proteobacteria* that include *Vibrio* species.

Figure 1 shows bacilli-like structures attached to an intestine fragment taken at the end of the assay from a fish fed with SPC30Pro diet.

To know the modulation effect of *B. subtilis* 9b in *T. macdonaldi* intestinal microbiota throughout the trial, fluorescent in situ hybridization was carried out in fish fed with SPC30Pro, SPC30C, and DCML (Figure 2). A low G + C Gram-positive bacteria increment is observed in fish fed SPC30Pro compared to DCML and SPC30C. However, in SPC30Pro treatment *γ-Proteobacteria* decreased in number at the end of the trial. On the other hand, treatment with SPC30C without *B. subtilis* could not reduce the number of *γ-Proteobacteria* at the end of the experiment as SPC30Pro did. DCML commercial diets showed the lowest bacterial numbers at the end of the feeding trial (Figure 2).

Intestinal microbiota modulation by *B. subtilis* 9b in *T. macdonaldi* fed with SPC30Pro diet can be observed in Figure 3. An increase in low G + C Gram-positive bacterial number can be observed at the end of the feeding trial (Figure 3b,c). However, *γ-Proteobacteria* number was reduced in the same treatment (Figure 3e,f). Thus, *B. subtilis* 9b can modulate intestinal microbiota of *T. macdonaldi*, which could induce beneficial effects observed in totoaba fed SPC30Pro treatment (Table 2 and Table 3).

### 3.4. T. macdonaldi Behavior after V. harveyi Infection

A total of six fish from DCML, SPC30C, and SPC30Pro treatments were infected with *V. harveyi* to evaluate *T. macdonaldi* behavior after pathogen inoculation. In the first two days, all fish showed reduced and erratic swimming, however, after the third day, fish fed with SPC30Pro showed a recovery, presenting more motility than the SPC30C control diet and DCML commercial diet (Table 4). Additionally, DCML and SPC30Pro treatments similarly improved feed intake after the third day; however, SPC30C showed delayed feed intake (Figure 4).

## 4. Discussion

*Bacillus* species have attracted great attention lately due to their capacity to improve growth performance and health status in aquacultured animals such as orange-spotted grouper (*Epinephelus coioides)* [21], shrimp (*Litopenaeus vannamei)* [22,23], bullfrog (*Lithobates catesbeianus*) [24], Nile tilapia (*Oreochromis niloticus*) [36,37], parrot fish (*Oplegnathus fasciatus*) [38], cobia (*Rachycentron canadum*) [39], Indian major carp (*Labeo rohita*) [40], red sea bream (*Pagrus Major*) [41], and sea bream (*Sparus aurata L*.) [42].

SBP addition in feed formulations of carnivorous species such as *T. macdonaldi* have already been studied. In this sense, Fuentes-Quesada and collaborators [30] found that substitutions of up to 23% of SBM induce adverse effects in fish. In addition, Trejo-Escamilla and collaborators [31] found that *T. macdonaldi* can tolerate up to 34.17% of SPC before developing adverse effects. Therefore, authors reported that *T. macdonaldi’s* low tolerance to SBP was mainly due to non-digestible carbohydrates and enzyme inhibitors present in SBM and SPC, respectively [30,31].

In this work, *T. macdonaldi* fed with SPC30Pro diet presented better growth performance and less visceral fat content than DCML commercial diet (Table 2 and Table 3). Thus, considering the health status of animals fed with SPC30Pro, this formulation seems to be an alternative to the Skretting© formulation used in this work (Table 3 and Table 4). Taking into account the larger visceral fat content found in animals fed with DCML diet, we speculate that the lipid concentration (12%) used to formulate this aquafeed could be responsible. In this sense, some reports indicate that high lipid concentration in aquafeeds could affect feed intake in carnivorous species [43]. High lipid concentration is principally used to feed cold water fish. Nevertheless, experiments with totoaba were developed at 25 °C; therefore, non-metabolized lipids were accumulated around the viscera. In addition, a 7% lipids concentration used in experimental diets was based on lipid content found in *T. macdonaldi* [44].

Fish fed with SPC60C and SPC60Pro presented lower growth and a decreased survival rate than SPC30C and SPC30Pro, respectively (Table 2). However, fish fed with SPC60Pro formulation showed 20% more survival and doubled weight gain than SPC60C without *B. subtilis* 9b strain; results that were not observed with SPC30C and SPC30Pro (Table 2). Therefore, we suggest that FM levels in SPC30Pro could be affecting *B. subtilis* behavior; maybe high levels of free nitrogen could be inhibiting *Bacillus* enzymes activity in SPC30Pro treatment [45,46]. In experiments developed with 100% of FM and *B. subtilis* 9b, *T. macdonaldi* presented less growth than its control without *B. subtilis* [32]. On the other hand, the increment of enzyme inhibitors in SPC60 formulations due to an increase in SPC could be affecting *T. macdonaldi* growth and survival (Table 2) [31]. These results suggest that *B. subtilis* 9b can degrade carbohydrates, lipids, and partially the proteins from SPC60Pro diet, however, it cannot eliminate all the enzyme inhibitors present in SPC60Pro formulation (Table 2). Thus, high levels of FM and SPC could be affecting *B. subtilis* 9b nitrogen homeostasis and *T. macdonaldi* proteases activity, respectively. Therefore, a new or complementary *Bacillus* strain with higher protease inhibitors degradation capacity must be found to obtain betters yields with 60% SPC substitutions. However, it is important to mention that SPC30Pro treatment produced the best results in this work.

Probiotics can affect muscle quality and the internal organs’ conditions. Some authors found that Pengze crucian carps (*Carassius auratus* var. Pengze) fed with formulations supplemented with *B. subtilis or B. cereus* produced a firmer, chewy, cohesive, and adhesive meat compared to fish fed with basal feed without *Bacillus* inclusion [47,48]. In this work, fish fed with SPC30Pro showed firmer and chewy fillets than most diets tested. In addition, fish fed with SPC30Pro showed bigger and larger organs compared with other treatments. These results suggest that *B. subtilis* 9b could be inducing benefits in *T. macdonaldi* muscle consistency and in organ health status (Table 3).

Probiotics also can modulate the host microbiota and, in this way, improve their health status. Totoaba autochthonous microbiota mainly comprises *Proteobacteria* from the *Vibrionaceae* family [49]. In this work, using FISH methodology, metabolically active γ-*Proteobacteria* and low G + C Gram-positive bacteria were evaluated in *T. macdonaldi* treated with SPC30Pro, SPC30C, and DCML diets (Figure 2). In this sense, *B. subtilis* 9b modulated *T. macdonaldi* intestinal microbiota by increasing low G + C Gram-positive bacteria amount and at the same time, it reduced the *γ-Proteobacteria* population (Figure 2). These results agree with results obtained by Tachibana and collaborators, who reported that *B. subtilis* and *B. licheniformis* added to Nile tilapia diets modulated intestinal microbiota; reducing *Proteobacteria* and increasing *Firmicutes* population [50]. However, DCML treatment reduced the bacterial number at the end of the experiment, indicating that DCML formulations contain compounds that inhibit the development of both groups (Figure 2).

*Bacillus* species produce antimicrobial peptides against many gram-positive and gram-negative pathogen bacteria [51]. These AMP can directly kill pathogens or induce the host immune system to respond against microbial infections. In this work, fish fed with SPC30Pro diet produced a faster recovery against *V. harveyi* infection. However, fish fed with DCML and SPC30C presented a slower recovery (Table 4 and Figure 4). Challenges with *B. subtilis* 9b and *V. harveyi* using agar plates with marine medium, showed that *B. subtilis* has the capacity to limit *V. harveyi* growth (data not shown). Therefore, higher resistance observed in fish fed with SPC30Pro could be related to in vitro antimicrobial capacity showed by *B. subtilis* 9b strain. *B. subtilis* 9b fish immune system stimulation requires more research before any assumption. However, *Bacillus* has been shown to boost different aquatic organisms’ immune system [22,38,52,53,54].

## 5. Conclusions

*T. macdonaldi* fed with SPC30CPro and SPC60CPro feeds containing high levels of SPC and *B. subtilis* 9b for ninety days improved growth performance and enhanced health status compared with SPC30C and SPC60C treatments without *B. subtilis* 9b. Therefore, *B. subtilis* 9b probiotic strain had the capacity to increase tolerance to ingredients such as corn starch and SPC present in these formulations. Furthermore, *B. subtilis* 9b showed the capacity to modulate intestinal microbiota increasing low G + C Gram positive bacteria and lowering *γ-Proteobacteria*, which seems to increase *V. harveyi* resistance. Moreover, muscle and organ characteristics were also improved in treatments containing *B. subtilis* 9b strain. Therefore, these results open the possibility to formulate functional feeds with alternative and economical vegetable ingredients and *B. subtilis* 9b, without generating animal health problems.

## Figures and Tables

**Figure 1 animals-12-03422-f001:**
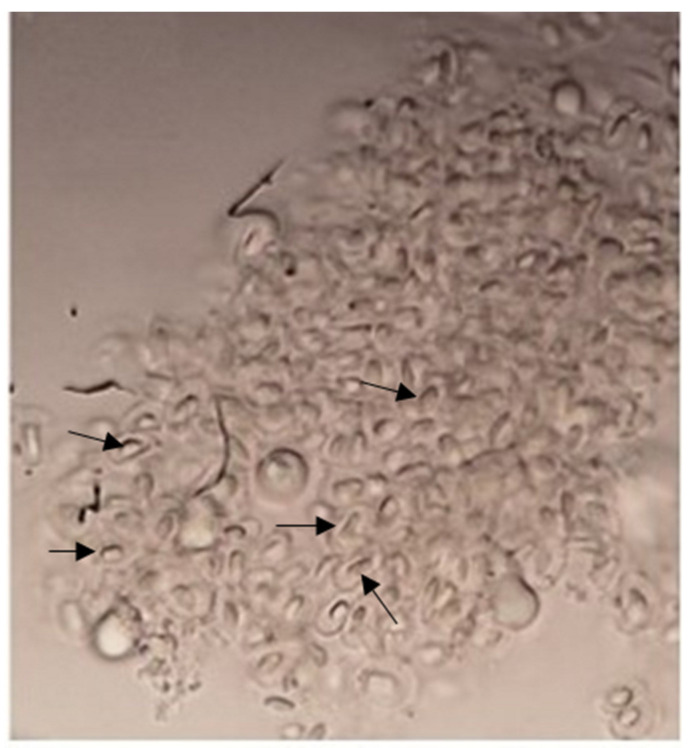
Black arrows point to some bacilli-like structures in fish intestine fed with SPC30Pro diet.

**Figure 2 animals-12-03422-f002:**
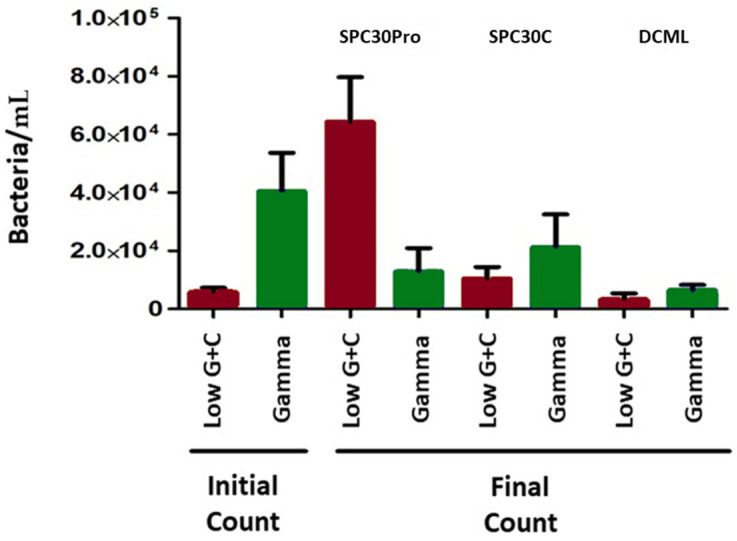
Bacterial count in intestinal samples of fish fed with SPC30Pro, SPC30C, and DCML at the beginning and end of the trial.

**Figure 3 animals-12-03422-f003:**
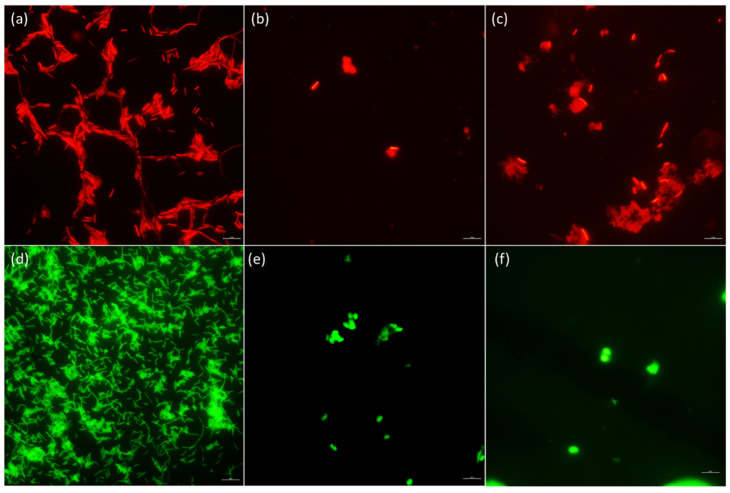
FISH of intestinal samples taken from totoaba fed with SPC30Pro diet at the beginning and end of the trial; (**a**) low G + C Gram-positive control strain (*Bacillus subtilis* 9b); (**b**) Low G + C Gram-positive bacteria from intestine samples at the beginning of the study; (**c**) Low G + C Gram-positive bacteria from intestine samples at the end of the study; (**d**) *γ-Proteobacteria* control strain (*Vibrio harveyi* ATCC14126); (**e**) *γ-Proteobacteria* from intestine samples at the beginning of the study; (**f**) *γ-Proteobacteria* from intestine samples at the end of the study.

**Figure 4 animals-12-03422-f004:**
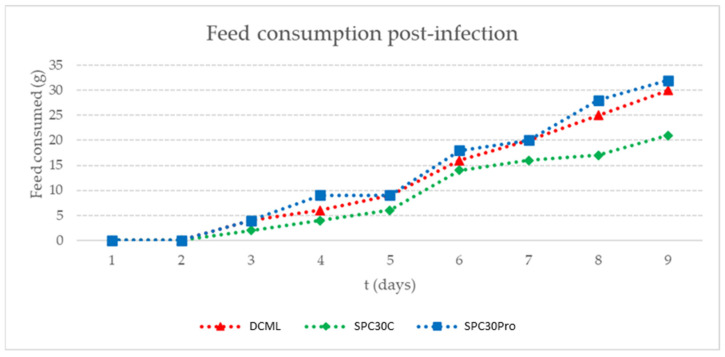
*Totoaba macdonaldi* feed intake behavior after *Vibrio harveyi* infection.

**Table 1 animals-12-03422-t001:** Ingredients (%) of feeds containing 30% and 60% of SPC substitution for juvenile *Totoaba macdonaldi*.

Ingredients	SPC30C	SPC30Pro	SPC60C	SPC60Pro
FM	42.05	42.05	25.31	25.31
SPC	20.26	20.26	39	39
Corn starch	17	17	15	15
Wheat flour	4	4	4	4
Corn flour	4	4	4	4
Gelatin	2.5	2.5	2.5	2.5
Fish Oil	3	3	3	3
Cellulose	2.1	2	2.1	2
Mineral mix	3.5	3.5	3.5	3.5
Vitamin mix	1.59	1.59	1.59	1.59
*B. subtilis* 9b	0	0.1	0	0.1

**Table 2 animals-12-03422-t002:** Growth performance of *Totoaba macdonaldi* fed with two levels of SPC and *Bacillus subtilis* 9b.

	DCML ^A^*	SPC30C ^B^*	SPC30Pro ^C^*	SPC60C ^D^*	SPC60Pro ^E^*
IW	152.6 ± 1.77	152 ± 1.9	153 ± 1.87	154.3 ± 2.068	153.1 ± 1.632
FW	618.4 ± 20.13 ^DE^	624.4 ± 16.97 ^DE^	645.4 ± 17.49 ^DE^	260.1 ± 57.65 ^ABCE^	406.2 ± 45 ^ABCD^
WG	306.1 ± 3.59 ^CDE^	312.5 ± 3.83 ^ED^	323.5 ± 3.86 ^ADE^	68.84 ± 0.94 ^ABCE^	165.9 ± 1.76 ^ABCD^
SGR	1.631 ± 0.03 ^DE^	1.637 ± 0.02 ^DE^	1.682 ± 0.02 ^DE^	0.360 ± 0.22 ^ABC^	1.002 ± 0.14 ^ABC^
TGC	1.508 ± 0.03 ^DE^	1.515 ± 0.03 ^DE^	1.567 ± 0.03 ^DE^	0.3445 ± 0.19 ^ABC^	0.9079 ± 0.13 ^ABC^
DFI	4.31	5.02	4.93	1.02	2.36
S (%)	100	100	100	61.11	83.33
IIW	4.05	4.11	4.22	1.69	2.65

*A (DCML), B (SPC30C), C (SP30Pro), D (SPC60C), and E (SPC60Pro) superscripts indicate a significant difference (*p* < 0.05); IW: initial weight; FW: final weight; WG: weight gained; SGR: specific growth rate; TGC: thermal growth coefficient; DFI: daily feed intake; S: survival; IIW: increase in initial weight.

**Table 3 animals-12-03422-t003:** Organs condition in fish fed with SPC treatments and *Bacillus subtilis* 9b strain.

Feed	Muscular Firmness ^1^	Visceral Fat ^1^	Organ’s Condition			
			Liver Color and Size (cm)	Spleen Size (cm)	Intestine Length (cm)	Bile
DCML	++	+++	Pink (12.7)	1.05	18.7	Yellow
SPC30C	+++	++	Pink (12.35)	1.57	22.47	Yellow
SPC30Pro	+++	++	Pink (12.68)	1.76	24.74	Greenish
SPC60C	+	++	Pale (11.16)	1.04	20.9	Yellow
SPC60Pro	++	++	Pale (10.4)	1.12	23.61	Yellow

^1^ +: low; ++: moderate; +++: high.

**Table 4 animals-12-03422-t004:** *Totoaba macdonaldi* motility after *Vibrio harveyi* infection.

Days Post-Infection	DCML	SPC30C	SPC30Pro
1	°	°	°
2	°	°	°
3	°	°°	°°°
4	°	°°	°°°
5	°°	°°	°°°
6	°°°	°°°	°°°
7	°°°	°°°	°°°
8	°°°	°°°	°°°
9	°°°	°°°	°°°

**°°°** High motility; **°°** Medium motility; **°** Low motility.

## Data Availability

Data is contained within the article.

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
