# Peer review of "Bacillus subtilis Effects on Growth Performance and Health Status of Totoaba macdonaldi Fed with High Levels of Soy Protein Concentrate"

_animals, 2022, doi:10.3390/ani12233422_

Round 1
Reviewer 1 Report
The authors contextualize well the problem and theorize solutions accordingly. The environmental problem of using soya could be better addressed, because, although is a solution to fish meal, if it comes from a non-certified area, it raises other problems. Furthermore, the lack of information about the composition of the control diet (DCML) prevents the authors to achieve better conclusions. Lastly, the microbiota methodology, results and discussion raised several concerns.
1. The statement in lines 29-31 can mislead the reader. Looks like diet with 60% substitution and B. subtilis (SPC60CPro) double their weight and survival compared to the control group (DCML). It is better explained in lines 362.
2. Line 83: missing species name in italic
3. Inclusion of dietary composition of the DCML diet would benefit the manuscript. At least macronutrient composition and their origin (animal or vegetable).
4. Not all analyses indicate how many fish were used.
5. Last sentence of section 3.2 (lines 262-264) is discussion. As well as, line 286 and line 314.
6. line 278: When the authors mention that in situ hybridization was used “to know the effect of B. subtilis 9b in T. macdonaldi intestinal microbiota throughout the trial”, it is a general analysis without a special motivation. In the reference #34 there is a purpose for targeting these bacterial groups.
It approach can, indeed, give some information about bacterial modulation, but the biological importance of these groups (role/function) is not mentioned or discussed. Other methodologies, such as 16S rRNA sequencing would give much more information about microbial modulation.
Still, the statement in line 286 “meaning that perhaps some antimicrobial compound was added to this formulation” is very speculative. The authors are analyzing two groups of bacteria only. Other bacterial might have the chance to thrive in the DCML group.
7. It is not presented any reasoning why groups with 60% replacement were not included in the analysis from the section 3.3 on.
8. Line 308: it was 2 or 6 fish per treatment? In figure 4, the feed consumption was per fish or per tank?
9. I am assuming that six fish per group were placed in separate tanks (one tank per group). However, results in section 3.4, figure 4, require some sort of statistical approach to be able to use terms such as “faster recovery” (line 311) or “increased feed intake after the third day” (line 313).
10. Table 4 requires a description in the results section.
11. In line 330, common names and genus of the species Epinephelus coioides and L. vannamei could be added for consistency reasons.
12. Lines 364-376: I believe the hypotheses presented in this paragraph are too ambitious. Please consider the following thoughts:
- Where did the authors based to conclude that higher levels of FM can affect the B. subtilis behavior? And, how much was the reduced growth of 100% FM with B. subtilis of the data not shown mentioned?
- One hypothesis for such reduced growth could be the fact that some energy could have been used by the immune system to respond to the presence of B. subtilis. But still, in this study, B. subtilis was beneficial in both 30% and 60% FM replacement.
- Regarding the lower growth performance in 60% replacement group, perhaps it was too much for the fish and B. subtilis was able to mitigate some of the effects of such inclusion.
- One problem of high inclusion could be intestinal inflammation. It is normally associated with intestinal shortening and it was not observed in the current study.
- Still, I believe that without knowing the composition of DCML is difficult to assess a proper conclusion regarding the effects of FM level on the B. subtilis behavior.
13. Line 382: general observation. What is "better organs condition" / “better-looking organs”? The author should develop such concepts in the material and methods. Is a bigger spleen better? What does it mean? What about the color of bile?
14. Line 412: Why nutrient absorption was never considered/discussed before in the manuscript?
Author Response
Dear reviewer, we appreciate a lot your time and effort to improve this manuscript, in this sense, the answers are placed after each question.
Thanks,
Prof. J. Olmos-Soto
- The statement in lines 29-31 can mislead the reader. Looks like diet with 60% substitution and subtilis (SPC60CPro) double their weight and survival compared to the control group (DCML). It is better explained in lines 362.
a. Done, sentence was improved. - Line 83: missing species name in italic
a. The species name has been corrected. - Inclusion of dietary composition of the DCML diet would benefit the manuscript. At least macronutrient composition and their origin (animal or vegetable).
a. Done, commercial diet formulation was included in materials and methods. - Not all analyses indicate how many fish were used.
a. Done, we added the number of fish used in the experiments. - Last sentence of section 3.2 (lines 262-264) is discussion. As well as, line 286 and line 314.
a. Comments were eliminated from results. - line 278: When the authors mention that in situ hybridization was used “to know the effect of subtilis 9b in T. macdonaldi intestinal microbiota throughout the trial”, it is a general analysis without a special motivation. In the reference #34 there is a purpose for targeting these bacterial groups.
a. Done, we added the purpose at the beginning of 3.3. - It approach can, indeed, give some information about bacterial modulation, but the biological importance of these groups (role/function) is not mentioned or discussed. Other methodologies, such as 16S rRNA sequencing would give much more information about microbial modulation. Still, the statement in line 286 “meaning that perhaps some antimicrobial compound was added to this formulation” is very speculative. The authors are analyzing two groups of bacteria only. Other bacterial might have the chance to thrive in the DCML group.
a. Done, comment was eliminated. - It is not presented any reasoning why groups with 60% replacement were not included in the analysis from the section 3.3 on.
a. Efforts were focus in SPC30 experiments because they produced the best growth performance and health status. - Line 308: it was 2 or 6 fish per treatment?
a. At the beginning of the 3.4 section is mentioned that these experiments were developed with 6 animals. - In figure 4, the feed consumption was per fish or per tank?
a. Per tank. - I am assuming that six fish per group were placed in separate tanks (one tank per group). However, results in section 3.4, figure 4, require some sort of statistical approach to be able to use terms such as “faster recovery” (line 311) or “increased feed intake after the third day” (line 313).
a. Movement behavior was recorded by observation and touching; it was easy to visualize the animals with slow movements after infection and when they began to have their normal behavior.
Feed consumption was measured every day and the first two days after infection the animals did not consume anything. However, after the third day they began to gradually increase their feed intake.
- Table 4 requires a description in the results section.
a. Done. - In line 330, common names and genus of the species Epinephelus coioidesand vannamei could be added for consistency reasons.
a. Done, common name and genus were added. - Lines 364-376: I believe the hypotheses presented in this paragraph are too ambitious. Please consider the following thoughts:
Where did the authors based to conclude that higher levels of FM can affect the B. subtilis behavior?
a. Two references were included about nitrogen homeostasis regulation. - And, how much was the reduced growth of 100% FM with B. subtilis of the data not shown mentioned?
a. Reference was included with respect that experiment. - One hypothesis for such reduced growth could be the fact that some energy could have been used by the immune system to respond to the presence of B. subtilis. But still, in this study, B. subtiliswas beneficial in both 30% and 60% FM replacement.
a. Not apply, because in in this study, B. subtilis was beneficial in both 30% and 60% FM replacement. - Regarding the lower growth performance in 60% replacement group, perhaps it was too much for the fish and B. subtiliswas able to mitigate some of the effects of such inclusion.
a. I agree with this idea, check the manuscript please. - One problem of high inclusion could be intestinal inflammation. It is normally associated with intestinal shortening and it was not observed in the current study.
a. Please check Table 3; SPC60C results show this treatment present the smaller intestine and the higher mortality. - Still, I believe that without knowing the composition of DCML is difficult to assess a proper conclusion regarding the effects of FM level on the B. subtilisbehavior.
a. The formulation of DCML was included. - Line 382: general observation. What is "better organs condition" / “better-looking organs”? The author should develop such concepts in the material and methods. Is a bigger spleen better? What does it mean? What about the color of bile?
a. Done, text was modified in discussion section. - Line 412: Why nutrient absorption was never considered/discussed before in the manuscript?
a. Done, text was modified.

Reviewer 2 Report
This study investigated the effects of dietary Bacillus subtilis on the growth performance and health status of Totoaba macdonaldi. The topical of this study is practical and important for the fish feed industry. Overall, the authors have obtained very interesting results. My only concern with this manuscript is about the “Statistical analysis.” In the design of this study, 30% and 60% of SPC substitution were used, and the “B. subtilis 9b strain” was used or not used in the feed. Are two parameters or one parameter in your statical analysis? To make it clear, the authors are suggested to revise their “Statistical analysis” to clarify the method further.
Author Response
Dear reviewer, we appreciate a lot your time and effort to improve this manuscript, in this sense, the answers are placed after each question.
Thanks,
Prof. J. Olmos-Soto
- This study investigated the effects of dietary Bacillus subtilis on the growth performance and health status of Totoaba macdonaldi. The topical of this study is practical and important for the fish feed industry. Overall, the authors have obtained very interesting results. My only concern with this manuscript is about the “Statistical analysis.” In the design of this study, 30% and 60% of SPC substitution were used, and the “ subtilis 9b strain” was used or not used in the feed. Are two parameters or one parameter in your statical analysis? To make it clear, the authors are suggested to revise their “Statistical analysis” to clarify the method further.
a. The analysis was done to compare the substitution with their respective treatment with added probiotics, therefore we took them as a single parameter for the statistical analysis.

Reviewer 3 Report
This paper seeks to understand whether different concentrations of the bacterium Bacillus subtilis added to fish feeds would influence weight gain and health status in the highly aquacultured and commercial fish Totoaba macdonaldi.
I believe the paper has good potential, but it still needs work.
General comments:
The introduction lacks a solid background on the major topics dealt with in the paper. I suggest that the authors review and integrate more information, also considering the audience of the chosen journal. See this recent paper for example:
EroldoÄŸan, Orhan Tufan, et al. "From the sea to aquafeed: A perspective overview." Reviews in Aquaculture (2022).
The methodology is unclear in some areas, also, I think some conclusions are highly speculative because they are made on data that does not have sufficient samples or a solid statistical analysis.
The results present some text that belongs to the methodology and some results that are not mentioned in the methodology.
The discussion is mixed with themes that belong to the introduction and should be reviewed carefully when the methodology and results problems mentioned above are resolved.
See attached file for detailed comments on specific areas of the manuscript.

Author Response
This paper seeks to understand whether different concentrations of the bacterium Bacillus subtilis added to fish feeds would influence weight gain and health status in the highly aquacultured and commercial fish Totoaba macdonaldi.
I believe the paper has good potential, but it still needs work.
a. Actually the papers seeks to evaluate the probiotic capacity of B. subtilis 9b previously evaluated in other carnivorous species, to improve T. macdonaldi tolerance to different amounts of SPC (30% and 60%) and its effects in growth performance and health status.
General comments:
The introduction lacks a solid background on the major topics dealt with in the paper.
a. I am not agree with this comment
I suggest that the authors review and integrate more information, also considering the audience of the chosen journal. See this recent paper for example:
EroldoÄŸan, Orhan Tufan, et al. "From the sea to aquafeed: A perspective overview." Reviews in Aquaculture (2022).
The methodology is unclear in some areas, also, I think some conclusions are highly speculative because they are made on data that does not have sufficient samples or a solid statistical analysis.
The results present some text that belongs to the methodology and some results that are not mentioned in the methodology.
The discussion is mixed with themes that belong to the introduction and should be reviewed carefully when the methodology and results problems mentioned above are resolved.
- Review punctuation on line 39, please go into more detail and background about Aquafeed
a.Punctuation and aquafeed definition were added to the text. - On line 45 review “abundant” in this context
a. “abundant” was changed by “more accessible”. - On line 46 define functional feeds
a. A description of FF was added. - Line 61 At first mention of species within text the genus should be spelled out completely
a. Done. - Spell genus first on line 66
a. Done. - Define FDA acronym on line 70
a. Done. - Review grammar on line 74, 81
a. Done - Clarify the common name for T. macdonaldi
a. Done - Explain how the probiotic modulates the intestinal microbiota
a. Probiotics promote or restrict the growth of microorganisms present in the intestine by competition and/or by producing antimicrobial peptides. Therefore, we analyzed this modulation by Fluorescent in situ hybridization (FISH) and our results demonstrate that B. subtilis 9b is capable to stimulate low G+C Gram positive bacteria and low γ-proteobacteria numbers. - can you include detailed ingredient of this commercial feed as well?
a. Information about commercial feed ingredients was added. - not clear what these probiotics are, please specify. Would they interfere with B. subtilis strain?
a. Table 1 was modified to specify that “probiotics” added in diets is subtilis 9b. - 2.3 not clear why they were anesthetized… before taking their weight?
a. The phrasing of point 2.3 was modified.
- Are the standard equations in point 2.3 used in other papers as well? if so please cite the most relevant
a. References were added. - why do you number the equations? do you mention them somewhere in the text?
a. the numbering was removed. - ‘fillet texture’ not clear, please explain if this is a technical aquaculture term
a. The text was modified. - please detail the modifications or clarify if these are listed below in point 2.5
a. the modifications consisted in incubation times for control and sample preparation. - 5.3 not clear, please explain what kind of samples. How could the distal intestine be taken before the beginning of the experiment?
a. Samples were taken from five fishes previous the beginning of the study and from two fish from each treatment at the end of the experiment.
- This entire paragraph is unclear. What are samples and what are pellets?
a. Information of the samples was added. - 2. 7 Please clarify exactly on which data you performed statistical analyses, as it seems to be missing in some cases
a. Information about samples statistical analysis was added. - On line 237 review verb tenses
a. Done. - DFI missing from legend for table 2
a. DFI added to the legend. - 251 This should be part of the methodology
a. It was described on point 2.4 as organ´s condition. The text was modified to describe the organs that were analyzed. - 262 Using just three fish cannot count as a valid assessment of muscular condition
a. Why not? Whether differences in tissue condition were clear and consistent between treatments. In addition, some cited articles obtained the same results using Bacillus species. - Line 279 You didn’t do any statistical analysis to compare the results from different groups of feed?
a. FISH results are very conclusive between the treatments, therefore we are only showing the standard deviation. In addition, some cited articles obtained the same results using Bacillus species.
- Line 286 I think that it is important to find out about the formulation of DCML
a. Information about the formulation of DCML was added to the point 2.1. - How was motility measured? It’s not mentioned in the methodology
a. Movement behavior was recorded by observation and touching; it was easy to visualize the animals with slow movements after infection and when they began to have their normal behavior.
- Line 314 This is speculation because it’s only done on six fish
a. This line was eliminated from results, however fish behavior suggest that B. subtilis 9b improves T. macdonaldi tolerance to V. harveyi. In addition, FISH results demonstrate that B. subtilis 9b controlled γ-proteobacteria proliferation, here it is important to remember that V. harveyi belongs to this proteobacterial group. - Line 324-346 belong more to the introduction. I suggest to briefly summarize all the results in this first paragraph, and then you go on with the discussion.
a. Done, most of this information was eliminated.

Round 2
Reviewer 1 Report
The quality of the new version of the manuscript was increased considerably. The methodology and discussion/conclusion are more adequate.
Only fine comments:
1. I cannot find the diet on the supplier webpage. Perhaps it was discontinued.
2. In the formulas of DFI and Survival rate, it should say “fish” and not “fishes”.
3. In table 2, legend should say what “IIW” stands for.
Author Response
The quality of the new version of the manuscript was increased considerably. The methodology and discussion/conclusion are more adequate.
a. Thank you, we appreciate your comments.
Only fine comments:
- I cannot find the diet on the supplier webpage. Perhaps it was discontinued.
a. Product number 12143085 is for the Skretting Marine MX feed. In the word document we attach a picture of the feed. - In the formulas of DFI and Survival rate, it should say “fish” and not “fishes”.
a. wording was corrected. - In table 2, legend should say what “IIW” stands for.
a. Legend for IIW was added to table 2.
